# Using *Drosophila melanogaster* to Analyse the Human Paralogs of the ESCRT-III Core Component Shrub/CHMP4/Snf7 and Its Interactions with Members of the LGD/CC2D1 Family

**DOI:** 10.3390/ijms23147507

**Published:** 2022-07-06

**Authors:** Miriam Baeumers, Katharina Schulz, Thomas Klein

**Affiliations:** 1Institute of Genetics, Heinrich-Heine-Universitaet Duesseldorf, Universitaetsstr. 1, 40225 Duesseldorf, Germany; miriam.baeumers@hhu.de; 2Institut für Angewandte Bewegungswissenschaften, Professur für Sportmedizin/-Biologie, Technische Universität Chemnitz, Thüringer Weg 11, 09126 Chemnitz, Germany; katharina.schulz@hsw.tu-chemnitz.de

**Keywords:** ESCRT, ESCRT-III, CHMP4, Lgd, LGD1, LGD2, CC2D1A, CC2D1B, Notch pathway, endosomal pathway

## Abstract

The evolutionary conserved ESCRT-III complex is a device for membrane remodelling in various cellular processes, such as the formation of intraluminal vesicles (ILVs), cytokinesis, and membrane repair. The common theme of all these processes is the abscission of membrane away from the cytosol. At its heart in *Drosophila* is Shrub, CHMP4 in humans, which dynamically polymerises into filaments through electrostatic interactions among the protomers. For the full activity, Shrub/CHMP4 requires physical interaction with members of the Lgd protein family. This interaction is mediated by the odd-numbered DM14 domains of Lgd, which bind to the negative interaction surface of Shrub. While only one Lgd and one Shrub exist in the genome of *Drosophila*, mammals have two Lgd orthologs, LGD1/CC2D1B and LGD2/CC2D1A, as well as three CHMP4s in their genomes, CHMP4A, CHMP4B, and CHMP4C. The rationale for the diversification of the ESCRT components is not understood. We here use *Drosophila* as a model system to analyse the activity of the human orthologs of Shrub and Lgd at an organismal level. This enabled us to use the plethora of available techniques available for *Drosophila*. We present evidence that CHMP4B is the true ortholog of Shrub, while CHMP4A and CHMP4C have diverging activities. Nevertheless, CHMP4A and CHMP4C can enhance the activity of CHMP4B, raising the possibility that they can form heteropolymers in vivo. Our structure-function analysis of the LGD1 and LGD2 indicates that the C2 domain of the LGD proteins has a specific function beyond protein stability and subcellular localisation. Moreover, our data specify that CHMP4B interacts more efficiently with LGD1 than with LGD2.

## 1. Introduction

Membrane turnover is essential for the maintenance of the membrane compartments of the cell. The turnover is achieved by the exchange of membrane vesicles between compartments, which are formed via the abscission of membrane fragments from the donor compartment and consequently fuse with the acceptor compartment. In most cases, the exchange of membranes is accompanied by the transport of protein cargo. Abscission can occur from a compartment into the cytosol, e.g., during Clathrin-dependent and -independent endocytosis [1]. It can also occur in the reverse topological direction, away from the cytosol.

For this reverse abscission, only one mechanism is known, mediated by the Endosomal Sorting Complexes Required for Transport (ESCRT) machinery. It regulates the abscission of membrane patches in many processes, among them the formation of intraluminal vesicles (ILVs) at the limiting membrane (LM) of the maturing endosome (ME) [2,3,4]. The machinery consists of five in sequence acting complexes that 1. concentrate ubiquitylated membrane-inserted or -associated cargo, e.g., signalling receptors, at sites of ILV formation (ESCRT-0, ESCRT-I, and ESCRT-II) and 2. subsequently pinch off this membrane patch into the endosomal lumen as ILVs (ESCRT-III and Vps4) [5]. ESCRT-III consists of four core components and several accessory elements, which are all structurally related and belong to the CHMP protein family. They assemble into a complex only temporally at the LM. At its heart is CHMP4, Shrub in *Drosophila*, which polymerises at membranes into long filaments [5]. The polymerisation of Shrub/CHMP4 is initiated at the membrane by CHMP6/Vps20. The emerging polymer might serve as a template for the polymerisation of later acting ESCRT-III subunits, such as CHMP3/Vps24 and CHMP2/Vps2 [5]. With exception of Vps20/CHMP6, all ESCRT-III components cycle between the cytosol and the LM and exist in the cytosol in the monomeric closed conformation. Recently, it has been observed that the polymers are constantly remodelled by the AAA-ATPase Vps4, which is recruited via its binding to the MIM domains of the ESCRT-III subunits [6]. Hence, a given ESCRT-III subunit assembles at the LM of the ME only for a short time and is, therefore, hardly detectable on endosomes. Their transient endosome location is only clearly revealed if the turnover is stalled by the inactivation of Vps4.

We have recently found that in the polymer, Shrub protomers are extended helical hairpins that bind each other via electrostatic interactions of two complementary charged surfaces in a staggered array [7]. A similar staggered polymerisation was found for the yeast ortholog Snf7. Surprisingly, Snf7 protomers bind each other largely via hydrophobic interactions [8]. For the polymerisation to happen, Shrub and Snf7 appear to change from the closed to the open conformation. While this open conformation in the polymer appears to be characteristic for the CHMP4/Shrub/Snf7 orthologs, other CHMP family members appear to polymerise into different conformations, ranging from closed to semi-open [5]. Moreover, lateral interactions between different polymers have also been recently observed. It is currently not understood how these described filaments achieve membrane abscission.

In *Drosophila*, the loss of function of ESCRT-I, -II, and -III causes the ectopic activation of the Notch pathway, overgrowth of imaginal discs, and the loss of the epithelial polarity [9].

The *Drosophila* tumour-suppressor Lethal (2) giant discs (Lgd) has been recently identified as a main regulator of the activity of Shrub/CHMP4s [2]. Lgd is present in the genome of all metazoans and possesses four repeats of the family defining Drosophila melanogaster 14 (DM14) domain, followed by a phospholipid-binding C2 domain in its C-terminus. Mammals contain two variants, termed LGD1 and LGD2, also known as CC2D1B and CC2D1A, respectively. The structure/function analysis revealed that the DM14 domains are essential for function. The C2 domain of Lgd is required for the correct subcellular localisation and protein stability. A further specific function of the C2 domain could not be determined, because of the instability of the corresponding deletion variants [10].

Similar to loss of function of components of the ESCRT-I, -II, and -III, the loss of *lgd* function results in the uncontrolled ligand-independent activation of the Notch signalling in *Drosophila* [2]. This ectopic activation causes the over-proliferation of the disc cells [11]. In mammals, LGD proteins were shown to be involved in several different processes, besides the regulation of the ESCRT machinery. They are involved in the regulation of MAPK and Toll-like receptor 4. In addition, CC2D1A was shown to be involved in the regulation of multiple signalling pathways, including AKT, NF-kB, and protein kinase A [2]. Recent work revealed that CC2D1B is involved in the closure of the nuclear envelope after mitosis [12].

It was previously found that human LGD1 and LGD2 can functionally replace Lgd in *Drosophila* [13]. Moreover, it has been shown that, similar to Shrub, CC2D1A and B can bind to CHMP4B in vitro and in cell culture cells [4,13]. These results suggest that the function of the Lgd/CC2D1 family is conserved throughout evolution and that they function in a partially redundant manner [13]. The functional redundancy was recently confirmed by the genetic analysis of mouse double mutants [14]. Mutations in *LGD2/CCD1A* cause an autosomal inherited form of mental retardation and autism syndrome in humans [15,16].

While Lgd has four DM14 domains, only the odd-numbered domains of Lgd bind Shrub and are essential for its function [7]. They are helical hairpins with a positively charged surface, which bind to the negative interaction surface of Shrub, which is also required for interaction among Shrub protomers during polymerisation. Hence, the binding of Shrub to Lgd and to another Shrub protomer is mutually exclusive. Cell culture experiments initially suggested that Lgd is a negative regulator of Shrub [2,4]. However, in vivo analysis in *Drosophila* subsequently showed that Lgd is a positive regulator required for the full activity of Shrub, despite the mutually exclusive binding [3,10]. This is demonstrated by the finding that the *lgd* mutant phenotype can be rescued by simply increasing the levels of Shrub activity [3]. This finding also indicates that, in *Drosophila*, Lgd appears to have no function other than positively regulating the activity of Shrub.

While only one Lgd and one Shrub exist in the genome of *Drosophila*, mammals have two Lgd orthologs, LGD1/CC2D1B and LGD2/CC2D1A, as well as three CHMP4s in their genomes, CHMP4A, CHMP4B, and CHMP4C [2,5]. The rationale for the diversification of the different ESCRT-III and Lgd factors is not fully understood. Most is known about CHMP4B, which is involved in almost all ESCRT processes studied. CHMP4C is a regulator of the mitotic spindle checkpoint [17], while no specific cellular function is so far known for CHMP4A. It is also not known whether the CHMP4s provide redundant function, whether they can form heteropolymers, and whether they preferentially interact with one of the two LGDs. Furthermore, it is not shown that the discovered electrostatic interactions between Shrub protomers in the ESCRT-III filament and the interactions of Shrub with Lgd found for *Drosophila* are conserved in mammals and therefore of general meaning.

To answer these questions, we here investigated the function of the human orthologs of Shrub and Lgd in *Drosophila* to be able to use the plethora of techniques available in this model system for analysis. Our data indicate that the basic principles of the activity of Shrub and its interaction with Lgd are conserved. We present the structure function analysis of the LGD1 and LGD2 that indicates that the C2 domain of the LGD proteins has a specific function beyond protein stability and subcellular localisation. We provide evidence that the CHMP4B is the true functional ortholog of Shrub and that CHMP4A and C have partially divergent functions. Our data also raise the possibility that CHMP4B can form functional heteropolymeric filaments with CHMP4A and CHMP4C in vivo. In addition, our results suggest that CHMP4B interacts more efficiently with LGD1 than with LGD2.

## 2. Results

Several differently tagged versions of CHMP4B and Shrub have been used in the past for a multitude of experiments, without exactly knowing their activity. Research in the last decade revealed that important sequences for function are located at the extreme N- and C-terminus of Snf7 and Shrub [18,19,20]. The addition of tags might interfere with the function of these regions. The C-terminus harbours the MIM domain that is required for the interaction with Vps4 during disassembly of the Shrub polymer (e.g., [21]). It has been previously shown that the addition of GFP to the C-terminus inactivates Shrub, indicating that the addition of at least large tags to the C-terminus is problematic [19]. To avoid any problems with tagging in our experiments here, we tested the functionality of various tagged variants of Shrub in a rescue assay in *Drosophila* (see Appendix A). The expression of the variants was controlled by a genomic fragment encompassing 510 bp up- and downstream of the *shrub* ORF (*shrubP*). If *shrubP* controls the expression of full-length untagged Shrub, a strong rescue of the embryonic lethal *shrub* mutant flies is observed [7]. Most of the rescued flies die as pharate adults, but a few escapers even hatch. The presence of two copies leads to a complete rescue (Appendix A). Thus, *shrubP* encompasses most of the regulatory promoter sequences and is suitable for the analysis of the activity of the generated variants.

In agreement with the incompatibility of C-terminal tagging with function, we found here that a variant with a C-terminal Myc tag fails to rescue *shrub* mutants, indicating that it is not functional. Thus, the addition of even a small tag to the C-terminus appears to be deleterious. Therefore, we concentrated on tagging the N-terminus (Appendix A).

In a first round, we generated variants where GFP is fused to the N-terminus of Shrub, either via a short poly-glycine linker or a long flexible LAP-tag linker. In addition, we generated V5-, Myc- and HA-tagged variants. In contrast to untagged Shrub, both N-terminally GFP-tagged variants had no rescue activities, even if present in two copies. The *shrub* mutant flies died in their presence before the third larval instar stage (L3). Thus, a GFP linked to Shrub even via the long and flexible LAP-tag linker is not functional, if present alone in the genome. This is in agreement with the analysis of a correspondent construct in yeast [18]. In contrast, the variants with one of the short tags (Myc, V5, or HA) directly fused to the N-terminus rescued as good as the untagged version, indicating that they are active. Note that the HA tag must be fused directly to the N-terminus of Shrub, as we found that a variant where the HA tag was fused with a short glycine linker was inactive.

Altogether, our analysis indicates that short tags directly added to the N-terminus are tolerable, but the addition of tags to the C-terminus via a linker, as well as C-terminal fusions, is not functional.

### 2.1. Rescue Abilities of the Mammalian CHMP4s in Drosophila

Mammals have three orthologs of Shrub, CHMP4A, CHMP4B, and CHMP4C. We have recently shown that a BAC containing the *shrub* gene (*BAC^shrub^*) can fully rescue the *shrub* null mutant phenotype, even if present in only one copy [3]. When the ORF of *shrub* was replaced by that of CHMP4B (*BAC^shrub^-CHMP4B*), we only achieved full rescue if two copies were present (Figure 1 [3]). This indicated that CHMP4B can replace the function of Shrub to a large extent. We here found that similar constructs where the ORF of *shrub* was replaced by CHMP4A or CHMP4C, *BAC^shrub^-CHMP4A* and *BAC^shrub^-CHMP4C*, failed to rescue the *shrub* mutant phenotype, even if present in two copies (Figure 1). These results indicate that CHMP4B is the functional ortholog of Shrub, while CHMP4A and CHMP4C are paralogs with diverging activities.

To test whether the CHMP4s can be recruited to the endosomal membrane, we depleted the posterior half of the wing disc for Vps4 by expressing an UAS vps4-RNAi construct for 29 h with a combination of *en*Gal4 and *tub*Gal80^ts^. Loss of the depolymerising Vps4 leads to the accumulation of Shrub at the endosomal membrane [22]. The N-terminal Myc- or HA-tag allowed the determination of the subcellular localisation of the CHMP4s. We found that all three orthologs accumulate at the endosomal membrane of the ME as a result of the depletion of *vps4* function (Appendix A). Hence, the recruitment can be separated from the function of a given CHMP4: although not able to rescue the *shrub* mutant phenotype, CHMP4A and CHMP4C are able to be recruited to the endosomal membrane in *Drosophila*.

### 2.2. Analysis of CHMP4B Protomer Interaction

Previously, we have characterised the binding of Shrub protomers within the polymer and identified amino acids (AAs), which are responsible for their electrostatic interactions ([7], Figure 2A). We wondered whether these interactions are evolutionary conserved. A comparison of the AA sequence of Shrub with CHMP4B revealed that the crucial AAs for protomer interaction are conserved (Figure 2A). We generated the corresponding CHMP4B variants where individual AAs that were identified as important for function in Shrub are exchanged. The constructs were expressed under the control of *shrubP* and tested whether they can rescue *shrub* mutants to the same extent as CHMP4B (Figure 2A,B).

In summary, this study showed that the AAs identified for Shrub protomer interactions are also important for the activity of CHMP4B, with one exception (Figure 2B). The charge reversal at position R63 (R63E) leads to a rescue already in one copy, while the corresponding Shrub variant cannot rescue. However, in two copies, the *shrub R59E* leads to a better rescue than the corresponding *CHMP4B R63E* variant. The reason for the small discrepancies between Shrub and CHMP4B if only one copy is present is not clear at the moment. Note that in the case of the positively charged AAs, a similar charged neighbour is present (Figure 2A). It is possible that this second positively charged AA can somehow compensate the charge reversal at the other position and take over to a certain extent.

To gain further confirmation that the investigated AAs are important for the polymerisation of Shrub in vivo, we tested whether the charge reversal at one position can be compensated by a complementary reversal at the position of the interacting AA in the complementary electrostatic surface. The D at position 79 of the negatively charged surface contacts the R at position 59 of the positively charged surface of the next Shrub protomer ([7], Figure 2A). R59 corresponds to R63 in CHMP4B (Figure 2A). We found that the combination of Shrub D79K, which does not rescue (even with two copies, Figure 2B), with CHMP4B R63E, which in one copy rescues to the early pupal stage, leading to a better rescue to the pharate adult stage (Figure 2C). This allelic complementation suggests that Shrub can interact with CHMP4B via similar interactions, as it does with another Shrub protomer. Hence, it supports the notion that the interactions between the protomers in ESCRT-III filaments are conserved in metazoans. 

Previous results indicated that while the binding between protomers of the Shrub/CHMP4 polymer are based on electrostatic interactions, binding between the protomers of the Shrub/CHMP4 yeast ortholog Snf7 relies mostly on hydrophobic interactions [7,8]. This is supported by the lack of conservation of several of the AAs in the surfaces of Snf7 that are crucial for the interactions between Shrub or CHMP4 protomers ([7], Figure 2A). In agreement with these differences, we found that Snf7 were not able to rescue *shrub* mutants if expressed under control of *BAC^shrub^* (*BAC^shrub^*-*snf7*) (Appendix A).

In yeast, a variant of the Shrub ortholog Snf7 where GFP is linked to the N-terminus via a long flexible LAP-tag can be incorporated into an ESCRT-III filament and is functional in combination with untagged Snf7, although inactive if it is present alone [18]. Likewise, we here found that eGFP-LAP-Shrub, which were constructed in a comparable manner, was also inactive if solely present in the genome (Appendix A). To test whether it can provide some activity in combination with a functional Shrub/CHMP4 variant, we tested whether the addition of GFP-LAP-Shrub can improve the incomplete rescue of *shrub* mutants achieved by the presence of one copy of CHMP4B. We, indeed, found that the combination of *BAC^shrub^-CHMP4B* with *shrubP*-*eGFP-LAP-shrub* slightly improves the rescue of the mutants. The flies developed to the late instead of the early pupal stage and the ectopic activation of the Notch pathway was reduced (Appendix A). This result indicates that GFP-LAP-Shrub is weakly active in combination with a functional Shrub/CHMP4 family member. However, GFP-LAP-Shrub, which is located in the cytosol in wildtype cells, accumulated at the LM of endosomes in the rescued cells, indicating that it cannot efficiently be removed from the endosome (Appendix A). Thus, its function is clearly strongly reduced, even in combination with a functional CHMP4Bs.

### 2.3. Structure-Function Analysis of the LGDs in Drosophila

A systematic structure function analysis of LGD1 and LGD2 in vivo has so far not been performed. We used the previously described *lgd* rescue assay to carry out this analysis in *Drosophila* to evaluate the importance of the DM14 and C2 domains of the human orthologs in an in vivo situation [3]. All constructs were HA-tagged, controlled by the *lgd* promoter (*lgd*P) and inserted in the same genomic landing site to guarantee comparable endogenous expression [10]. Western blot analysis confirmed similar expression levels of Lgd-HA, LGD1-HA, and LGD2-HA in third instar larvae (Appendix A). 

We tested the rescue abilities of the LGD1 variants in the *lgd* null mutant and in a sensitised background where one functional copy of *shrub* was removed in addition (*lgd shrub/lgd +* genotype, [10,23]). Loss of *lgd* function results in over-proliferation of imaginal discs and extensive ectopic expression of Notch target genes, e.g., Wg, in a ligand-independent manner ([11]; Figure 3B, arrow). The mutant flies die during the early pupal stage, while flies of the sensitised background die earlier, at the beginning of the third larval instar stage and have only small wing imaginal discs ([10], Figure 3D). As reported previously, LGD2 only partially rescued *lgd* mutants (Figure 3F,F’, [13]). In contrast, we here found that LGD1 rescued the sensitised background to the adult stage, as previously found for Lgd (Figure 3E). The LGD1 rescued flies exhibited no obvious defects, but were sterile (both sexes). Since LGD1 rescued *lgd* mutants completely, the sterility of the rescued flies with the sensitised background suggests that also LGD1 requires the activity of Shrub for its function in *Drosophila*.

We previously found that Lgd-HA under control of the endogenous promoter is expressed at levels too low to be unambiguously detected in anti-HA antibody staining of wing imaginal disc [10]. Therefore, we over-expressed the human variants also with the Gal4 system to be able to determine their subcellular localisation. We found that LGD1 and LGD2 were located in the cytosol of imaginal disc cells, as has been previously found for Lgd (Appendix A, [24]). Thus, it is likely that, in *Drosophila*, LGD1/CC2D1B and LGD2/CC2D1A act in the cytosol and do not have an additional role in the nucleus, as has been proposed based on mammalian cell culture experiments [10,25].

Variants of LGD1 and LGD2, lacking either all DM14 domains, or the C2 domain failed to rescue *lgd* mutants, indicating that these domains are essential for function (Figure 3H–K). In *Drosophila*, the loss of the C2 domain in Lgd resulted in its mislocation into the nucleus and a dramatic reduction of its protein level [10]. This behaviour was not observed for LGD1ΔC2, which was more stable (Appendix A) and located in the cytosol (Appendix A). The inactivity of LGD1ΔC2, combined with its correct subcellular location, indicates that the C2 domain is required for a function beyond protein stability and subcellular localisation. We were not able to draw this conclusion from our previous experiments with Lgd, because of the low levels of LgdΔC2.

The sequence comparison between Lgd, LGD1 and LGD2 revealed that LGD2 has a longer tail after the C2 domain, a feature that is conserved in all mammalian species ([24,26], Appendix A). To test its importance in the fly, we generated a LGD2 variant where the region is deleted (LGD2Δ815). This variant rescued as good as LGD2 (Figure 3F–G’), indicating that it has no function in the Lgd-mediated processes conserved between the fly and humans.

Variants of LGD1 with only one even- and one odd-numbered DM14 domain (LGD1ΔDM14 1+2 and LGD1ΔDM14 3+4) were able to partially rescue the *lgd* mutant phenotype to the pharate adult stage (Figure 3L–M’ and Figure 4A–C’). The appearance of these flies was normal and the ectopic expression of Notch target genes, typically observed in *lgd* mutant wing imaginal discs, was reduced to only a few cells located close to the D/V boundary of the wing disc (Figure 3L’–M’, arrow). We previously found that, in the case of Lgd, the presence of only one odd-numbered DM14 is sufficient to completely rescue *lgd* mutants [7,10]. In agreement with this notion, we found that a variant where the first three DM14 domains were deleted fails to rescue (Figure 4D,D’). The analysis of LGD1 also suggest that, while both odd-numbered DM14 domains can provide substantial activity if present alone, for the full rescue both are required. 

The degree of rescue of the variants with only one odd-numbered DM14 domain was strongly reduced in the sensitised background (Figure 3N–O’). This synergistic phenotype confirms that LGD1 requires the function of Shrub in *Drosophila*, as was previously found for Lgd and also LGD2 [10,13].

### 2.4. LGD2^MR^ Is a Loss of Function Allele in Drosophila

Homozygosity of an allele of LGD2, here termed LGD2^MR^, causes an inheritable form of autosomal non-syndromic mental retardation in humans [15]. In this allele, a frameshift occurs after the third DM14 domain. The resulting protein comprises the first 408 aa of LGD2 fused to a 30 aa long C-terminal nonsense peptide and includes three of the four DM14 domains. Hence, it might be a gain of function allele that interacts with Shrub/CHMP4 in a non-constructive manner and therefore enhance the *lgd* mutant phenotype, e.g., as in the case of loss of one copy of *shrub*. Moreover, the fused nonsense peptide could add new function on the LGD2^mr^ variant. To test the nature of *LGD2^MR^*, we analysed its rescue abilities in *Drosophila*. We also generated corresponding *Lgd^MR^* and *LGD1^MR^* variants. All variants were inserted in the same genomic landing site and expressed with *lgdP*. We found that all variants were unable to rescue *lgd* mutant flies, indicating that they did not possess any detectable activity (Appendix A). Hence, *LGD2^MR^* is probably a strong loss of function allele. The associated mental retardation in humans is, therefore, probably caused by the loss of activity of *LGD2* and not by newly gained properties of fused nonsense peptide encoded by *LGD2^MR^*, or a dominant negative effect caused by the truncation of LGD2.

### 2.5. CHMP4B Collaborates with LGD1 in Drosophila

An important question is whether the LGDs are activators of CHMP4B, as suggested by the fly results, or inhibitors, as suggested by the mammalian cell culture experiments [3,4,10]. To do so, we asked whether the *shrb lgd* double mutant can be rescued by the combination of *BAC^shrub^-CHMP4B* with *lgdP-LGD1-HA*. Indeed, we found that this is the case. *lgd shrub* double mutants rescued with two copies of *BAC-shrub^CHMP4B^* die in the late second instar stage (L2) (Figure 5A,G). However, the addition of already one copy of LGD1 to this genotype results in a much better rescue, as the corresponding flies develop until the pharate adult stage (Figure 5B,B’,G). Hence, CHMP4B depends on the function of LGD1. This finding suggests that also the mammalian LGDs are activator of the function of CHMP4B in vivo.

### 2.6. The Interactions between LGD1 and CHMP4B Follow the Same Rules Found for Lgd and Shrub

To confirm the conservation of the interactions between Lgd and Shrub in mammals, we tested whether the AAs required for interaction identified in Lgd have the same importance for LGD1 and CHMP4B [23]. We tested the rescue abilities of two variants where either R412 or R416, which correspond to R389 and R393 in Lgd, respectively, were replaced by E. The changes were introduced into deletion variants that only contain DM14-3 and -4 to remove the observed functional redundancy of the odd-numbered DM14 domains (LGD1ΔDM14-1+2R412E and LGD1ΔDM14-1+2R416E). The activity of the variants was tested in the *lgd shrub* double mutant background in the presence of two copies of *BAC^shrub^-CHMP4B*, which allowed the development of the flies to the late second instar stage (Figure 5A). The additional presence of LGD1ΔDM14-1+2 causes an enhancement of the rescue of the *lgd shrub* double mutant, indicating that also the truncated LGD1 variant is able to interact with CHMP4B (Figure 5D,D’). In contrast, we found that the addition of LGD1ΔDM14-1+2R412E or LGD1ΔDM14-1+2R416E were not able to further rescue, indicating that the AA exchange resulted in loss of function of LGD1ΔDM14-1+2 (Figure 5E,F). These results suggest that the electrostatic interactions of LGD family members with CHMP4 members are evolutionary conserved. 

### 2.7. CHMP4B Interacts More Efficiently with LGD1 Than with LGD2

We have previously found that while the presence of already one copy *lgdP*-*LGD1* causes a complete rescue of *lgd* mutants, *lgdP*-LGD2 results only in a partial rescue, even if present in two copies ([13], Figure 3F,F’). The flies rescued by two copies of *lgdP*-*LGD2* died as pharate adults. 

We tested whether this incomplete rescue is due to the lack of its true interaction partner CHMP4B. To do so, we combined CHMP4B with LGD1 or LGD2 to test whether they can rescue *shrub lgd* double mutant flies. We found that two copies of CHMP4B in combination with two copies of LGD1 resulted in a full rescue of the *shrub lgd* double mutant flies, with the exception of the sterility also observed in *shrub* single mutant rescued by CHMP4B (Figure 5G). The combination of two copies of LGD2 and CHMP4B led to an incomplete rescue of *shrub lgd* double mutant flies to the pharate adult stage (Figure 5G). These findings indicate that CHMP4B is in functional relationship with both LGD paralogs, but can interact more efficiently with LGD1 than with LGD2. 

### 2.8. CHMP4A and CHMP4C Can Synergise with CHMP4B 

In contrast to CHMP4B, CHMP4A and CHMP4C were not able to rescue *shrub* mutants, indicating that they are not able to replace the function of Shrub. We wondered whether they can cooperate with CHMP4B, e.g., by forming heteropolymers. To do so, we performed two sets of experiments. First, we asked whether the addition of CHMP4C or A can improve the observed partial rescue of *shrub* mutants by CHMP4B (Figure 1A). We found that this is the case. The addition of each of the two CHMP4 paralogs led to an improved rescue, indicating that they can somehow cooperate with CHMP4B. The enhancement upon addition of CHMP4C was stronger than addition of CHMP4A, indicating that CHMP4B can more efficiently cooperate. Importantly, no rescue was observed if CHMP4B was replaced by CHMP4B^mut2^, a variant that is unable to interact with other CHMP4B protomers [4,7]. This suggests that the principles of the interaction of CHMP4B with itself and with CHMP4A or CHMP4C are similar and raises the possibility that the variants form heteropolymers to achieve the observed cooperativity.

In a second set of experiments, we asked whether we obtain similar results if the mammalian LGDs are present. For this purpose, we exploited the fact that one copy of *BAC^shrub^-CHMP4B* in the presence of two copies of *lgdP-LGD1* only partially rescues *shrub lgd* double mutants to the early third instar larval stage (Figure 6E). We asked whether the co-expression of *BAC^shrub^-CHMP4B* with *BAC^shrub^-CHMP4C*, or *BAC^shrub^-CHMP4A*, can improve the partial rescue of *shrub lgd* mutants achieved by one copy of *BAC^shrub^-CHMP4B*. We found that this is the case: The combination of *BAC^shrub^-CHMP4B* with *BAC^shrub^-CHMP4A* allowed the development to the late pupal stage, while the combination with *BAC^shrub^-CHMP4C* even to the pharate adult stage (Figure 6A–E). Altogether, these results indicate that CHMP4A and CHMP4C can provide ESCRT-III function in *Drosophila* only in combination with CHMP4B. Since neither CHMP4A or CHMP4C cannot provide any rescue in the absence of CHMP4B and the rescue improvement relies on the interaction between the CHMP4s, a likely explanation for the cooperativity between the CHMP4 variant is that they form heteropolymers with CHMP4B.

## 3. Discussion

The ESCRT-III complex is the only known device in the cell that can mediate the abscission of membranes away from the cytosol. The majority of the in vivo characterisation of ESCRT-III has been done in yeast. Many aspects of the function of the complex turned out to be evolutionary conserved and, therefore, of general importance. However, important differences between yeast and metazoans have also been discovered. For example, it turned out that the interaction between the protomers in an ESCRT-III filament are different: While the binding of Shrub protomers are based on complementary electrostatic interactions in *Drosophila*, it is based more on hydrophobic interactions in yeast [7,8]. Thus, although Shrub and Snf7 polymerise into a similar staggered array, the mechanism by which it is achieved differs. Consequently, the AAs identified required for interaction between Shrub protomers are also conserved in CHMP4s, but not in Snf7. In line with the differences, we found here that, in contrast to CHMP4B, Snf7 cannot rescue the *shrub* mutant phenotype. Another difference between yeast and metazoans is that the interaction partner of Shrub/CHMP4, the Lgd/CCD1 protein family is not present in yeast and other unicellular organisms. The interaction between Lgd/CC2D1 and Shrub/CHMP4 requires the negative electrostatic surface of Shrub/CHMP4 [23]. This surface is not present in Snf7, suggesting that it cannot interact with and does not require the activity of LGD proteins. In contrast, we here provide ample evidence that Shrub and also CHMP4B interact and require the activity of LGDs for their full function. This difference suggests that the change in the type of interaction among protomers in metazoans required the addition of a regulatory element, the Lgd family.

The ESCRT family is expanded in mammals in comparison to yeast and *Drosophila*. Three Shrub and two Lgd orthologs exist in mammals. It was not known whether the orthologs can provide similar functions, whether the CHMP4s can form heteropolymers, and whether a distinct LGD prefers a distinct CHMP4. We here characterised all three human orthologs of Shrub and Lgd in our humanised fly model to evaluate the general importance of the results we have previously found for the *Drosophila* orthologs and to address the raised questions. The results of this characterisation largely confirmed and also extended the previous findings obtained with the *Drosophila* orthologs, indicating that the mechanisms discovered in *Drosophila* are of general meaning, probably for all metazoans. The in vivo analysis allows to detect also very weak changes in activity, which would likely not be detected in the available cell culture assays. It also allowed the testing of the consequences of tagging. Our findings that the addition of small tags at the N-terminus does not disturb the Shrub activity should be taken in account in future experiments. We would like to mention that, while using the fly model is good as an improved cell culture system to investigate specific properties of human proteins, such as the verification/confirmation of functional interactions between two proteins and verification of important regions, the read out is indirect via the developing phenotypes and also probably does not include all aspects of functions mediated by the proteins in the human environment.

One important finding here is that CHMP4B is the true ortholog of Shrub and appears to form polymers based on similar electrostatic interactions among the protomers as in the case of Shrub. In agreement with this conclusion is that the AAs found to be important for polymerisation of Shrub are conserved in the CHMP4s. In addition, just like Shrub, CHMP4B interacts with the odd-numbered DM14 domains of the LGDs and requires their activity for function in *Drosophila*. Previous in vitro experiments with LGD2 already revealed the interaction of DM14-3 of LGD2/CC2D1A with CHMP4B [4]. We here identify DM14-1 and DM14-3 as redundant acting devices for interaction with CHMP4B. This has been previously also found for the odd-numbered DM14 domains of Lgd [23]. 

Our findings further support the notion that the LGDs are positive regulators of CHMP4B in vivo [3]. How can a factor that keeps Shrub/CHMP4 in the monomeric form enhance the activity of CHMP4, although it is active in a polymer? CHMP4 cycles between the cytosol, where it exists in the monomeric form, and the membrane, where it polymerises into a filament. One previously discussed possibility is that Lgd prevents the inappropriate polymerisation in the cytosol. Alternatively, it might take over Shrub/CHMP4 monomers at the membrane upon removal by Vps4 to prevent their incorporation in ILVs. In both scenarios Lgd would prevent the loss of net activity of Shrub/CHMP4. Our finding here that the membrane binding C2 domain of LGD1 is required for its function suggests that an association with the membrane is important and favours the second scenario. 

We found that the CHMP4A and CHMP4C cannot rescue *shrub* mutants, indicating that they are functionally different from CHMP4B and Shrub. One fundamental aspect that could not be resolved so far is whether the CHMP4s can form heteropolymers. We here found that CHMP4A and CHMP4C cannot rescue *shrub* mutants, but can do so to a limiting extend in combination of CHMP4B, indicating that the CHMP4 orthologs have also partial redundant functions. The partial rescue of the CHMP4B in combination with C or A depended on the presence of AAs in CHMP4B that are crucial for interaction among CHMP4B protomers. These findings are compatible with the formations of CHMP4 heteropolymers. It has been shown that the ESCRT-III filaments behave different in different membrane abscission processes: while the filaments exist for a considerably longer time during cytokinesis, they exist for only minutes during ILV formation [5,18]. The formation of heteropolymers is a possibility to generate filaments with different properties which better fulfil the requirements for a given process. An example is CHMP4C, which has a unique phosphorylation site required for the regulation of the polymerisation of ESCRT-III during cytokinesis [27]. However, there are alternative interpretations of our results. It has been proposed that the initial forming CHMP4B filament at the endosomal membrane serves as a template for later forming filaments, consisting of combinations of later acting CHMP family members, e.g., CHMP2 and CHMP3 [28]. Hence, although less likely in our opinion, it is possible that CHMP4B might serve as a template for the formation of later forming filaments consisting at least in part of the other CHMP4 paralogs.

ESCRT and Lgd proteins are involved in disorders of the brain, such as autism syndrome, mental retardation, and dementia [16,29]. In the case of LGD2, two alleles have been isolated that cause mental retardation and autism, but it was not clear whether they encode a dominant-negative version or have lost their function. The initial allele, here named LGD2^MR^, bears a frame shift and encodes a variant that includes the first three DM14-domains fused to a frameshift induced nonsense peptide of 30 aa. Since the odd-skipped DM14 domains that interact with CHMP4 are still present in this variant, it is possible that it can interact with CHMP4s and, as shown here, also with Shrub, and might act in a dominant-negative fashion. Moreover, it was not clear whether the fusion of the nonsense peptide added new properties to the LGD2^MR^ variant. However, we found that LGD2^MR^ is a loss of function allele in *Drosophila*. While we cannot exclude the possibility that in its normal environment it still acts in a dominant negative manner, it is at least not likely that it interacts with the CHMP4s in a deleterious manner. A more recent analysis of two other alleles of CC2D1A that encode even smaller proteins supports the notion that loss of function of CC2D1A causes the mental defects [16].

Altogether, our presented analysis of mammalian ESCRT component shows that the fly model can provide interesting and relevant results, despite the mentioned caveats.

## 4. Methods and Materials

### 4.1. Drosophila Genetics

The following fly stocks were used during this analysis. Mutants: *lgd^d7^* FRT40A [24], *shrub^4−1^ FRTG13* [19]. Notch activity reporter: Gbe+Su(H)-lacZ [30]. GAL4: *enGAL4* (Bloomington Drosophila Stock Center (BDSC)). Others: *lgdP-LGD1-HA* (*lgdP-CC2D1B-HA*) and *lgdP-LGD2-HA* (*lgdP-CC2D1A-HA*), [13]), *BAC^shrub^, BAC^shrub^-cDNA and BAC^shrub^-CHMP4B* [3], *UAS-lgd-HA* [24], shrubP-shrub [7]. This work: *BAC^shrub^-CHMP4A, BAC^shrub^-CHMP4C*, *BAC^shrub^-CHMP4B^mut2^*, *shrubP-Myc-CHMP4B*, *shrubP-HA-shrub*, *shrubP-Myc-shrub*, *shrubP-V5-shrub*, *shrubP-HA-3xGly-Shrub*, *shrubP-eGFP-3xGly-shrub*, *shrubP-eGFP-LAP-shrub*, *UAS-LGD1-HA*, *UAS-LGD2-HA*, *UAS-LGD1ΔC2-HA*, *UAS-LGD2ΔC2-HA*, *lgdP- LGD1ΔC2-HA*, *lgdP- LGD2ΔC2-HA*, *lgdP- LGD1ΔDM14-HA*, *lgdP- LGD2ΔDM14-HA*, *lgdP-LGD1ΔDM14-1+2-HA*, *lgdP-LGD1ΔDM14-1+2R412E*, *lgdP-LGD1ΔDM14-1+2R416E*, *lgdP-LGD1ΔDM14-3+4-HA*, *lgdP-LGD1ΔDM14-1-3-HA*, *lgdP-LGD2Δ815-HA*, *lgdP-HA-lgd^MR^*, *lgdP-HA-LGD1^MR^*, *lgdP-HA-LGD2^MR^*.

### 4.2. GAL4/UAS System to Drive Expression in Drosophila

UAS *lgd* constructs were generated by SOE-PCR (*Gene splicing* by *overlap extension)* [31] using pUAST-*lgdHA* [24] as a template. Amplified sequences were cloned into pUAST using *Not*I and *Kpn*I [32]. Primer sequences are available from the authors upon request.

### 4.3. Endogenous Expression in Drosophila

#### 4.3.1. *BAC^shrub^* Based Expression System

The transcription unit of *BAC^shrub^* was replaced by a cDNA of CHMP4A [265 aa, NP_054888.2 (*BAC^shrub^-CHMP4A*)] or CHMP4C [NP_689497.1 (*BAC^shrub^-CHMP4C*)] by performing recombination-mediated genetic engineering [33]. All constructs were inserted into the genomic attP landing site attP86Fb [34]. Primer sequences are available upon request.

#### 4.3.2. *Lgd* Rescue Assay (*lgd*P)

Lgd rescue constructs (lgdP) are based on constructs described previously [10,13,35]. All deletion constructs were generated by SOE-PCR [31] or Gibson Assembly [36] and flanked by the proximal *lgd* genomic elements (548 bp upstream and 553 bp downstream of the *lgd* ORF). Primer sequences are available upon request.

####  4.3.3. *shrubP*-Expression System

To test the functionality of several tagged Shrub variants, we used our previously published expression system (*shrubP*) [7]. Therefore, shrub cDNA (or CHMP4B cDNA for *shrubP-Myc-CHMP4B*) was tagged and flanked by the proximal *shrub* genomic elements (510 bp upstream and downstream of start and stop-Codon, respectively). For N-terminal tagged V5-, Myc-, HA-, eGFP-Shrub, and C-terminal tagged Shrub-Myc no flexible linker between Tag and Shrub has been used. For eGFP-LAP-Shrub a long flexible linker has been utilised [37]. In case of HA-3xGly-Shrub and eGFP-3xGly-Shrub, the linker consists out of three glycines. Furthermore, we performed site-directed mutagenesis using *shrubP-Myc-CHMP4B* and *shrubP-Myc-shrub* to generate the indicated charge reversal mutations (Figure 2). All *shrubP* constructs were inserted into the genomic attP landing site 86Fb [34].

### 4.4. Immunohistochemistry and Microscopy

Dissected wing imaginal discs were fixed with 4% paraformaldehyde (in PBS) for 30 min, washed in PBS, and permeabilised in 5%NGS in 0.3% Triton X-100 (in PBS) for 30 min at room temperature. Discs were incubated with primary antibody for 90 min (5%NGS in 0.3% PBT), washed with PBT, and stained with the corresponding secondary antibody (5%NGS in 0.3% PBT). The following antibodies were used: mouse anti-Wg 4D4 (1:50, Developmental Studies Hybridoma Bank (DSHB), Iowa City, IA, USA), mouse anti-NECD (Notch extracellular domain) C458.2H (1:100, DHSB), anti-HA C29F4 (1:2000, Cell Signaling Technology, Topsfield, MA, USA) or anti-HA 3F10 (1:3000, Roche, Basel, Switzerland), and rabbit c-Myc ab9106 (1:1500 abcam). 

Images were acquired at a Zeiss AxioImager Z1 using the Zeiss Apotome device.

### 4.5. Western Blot

To determine the expression level of the *Lgd/LGD* constructs, third instar larvae were dissected in ice-cold PBS. They were homogenised in lysis buffer (10% glycerol, 50 mM HEPES pH 7.5, 150 mM NaCl, 0.5% Triton X-100, 1.5 mM MgCl_2_, 1 mM EGTA and 4 μL/mL protease inhibitor cocktail (Sigma, Darmstadt, Germany)). After 15 min of incubation on ice with lysis buffer, the lysates were centrifuged for 15 min at 4 °C. In case of Snf7, dissected wing imaginal discs were incubated in Laemmli-buffer (250 mM Tris-HCl, 8% SDS, 40% glycerol, 8% beta-mercaptoethanol, 0.02% bromophenol blue) at 95 °C for 10 min. Protein lysates were loaded onto 10% SDS-PAGE gel and immunoblotted according to standard protocols. After blocking (5% milk powder in PBS), the blots were stained using the following antibodies (diluted in 2% milk powder in PBS): anti-HA (1:3000; 3F10, Roche), anti-Snf7 (1:5000) [38], and HRP-conjugated secondary antibodies (1:5000 Jackson Immuno Research or Dianova).

## Figures and Tables

**Figure 1 ijms-23-07507-f001:**
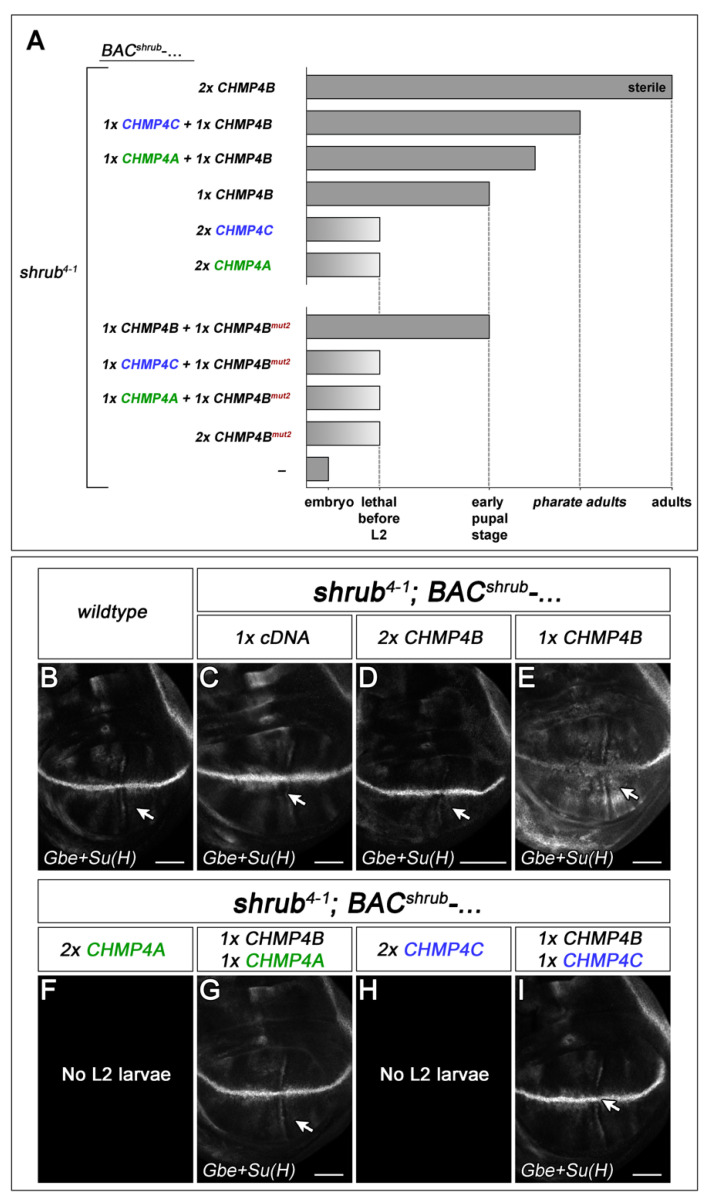
Functionality of human CHMP4 proteins in *Drosophila*. (**A**) Summary of the rescue experiments of CHMP4 orthologs. In contrast to CHMP4B, CHMP4A and CHMP4C were not able to rescue the shrub mutant phenotype. Nevertheless, they can improve the rescue of one copy of CHMP4B. The here-depicted CHMP4^mut2^ is a variant bearing three charge reversal mutations (E90R, E94R, and E97R) in its negatively charged interaction surface that abolish the interactions between CHMP4B protomers [4]. (**B**–**I**) Expression of the Notch activity reporter Gbe+Su(H) in *shrub* mutant disc rescued by the CHMP4 orthologs. In contrast to the control construct *BAC^shrub^-cDNA* (**C**), one copy of *BAC^shrub^-CHMP4B* (**E**) only partially rescues the *shrub* mutant phenotype, indicated by the weak ectopic expression of Gbe+Su(H) highlighted by the arrow in (**E**). In the case of *BAC^shrub^-cDNA* the transcription unit of *BAC^shrub^* is replaced by the cDNA of *shrub* instead of the cDNA of one of the CHMP4 orthologs [3]. (**D**) Two copies of *CHMP4B* result in a complete rescue, indicated by the pattern of Gbe+Su(H) (compare with B, wildtype). (**F**,**H**) Even in two copies CHMP4A and CHMP4C were not able to rescue the *shrub* mutant phenotype. (**G**,**I**) However, they improve the partial rescue by one copy of CHMP4B leading to a normalisation of the Notch activity. Scale bars (**B**–**I**) 200 µm. At least ten wing imaginal discs were analysed for each genotype.

**Figure 2 ijms-23-07507-f002:**
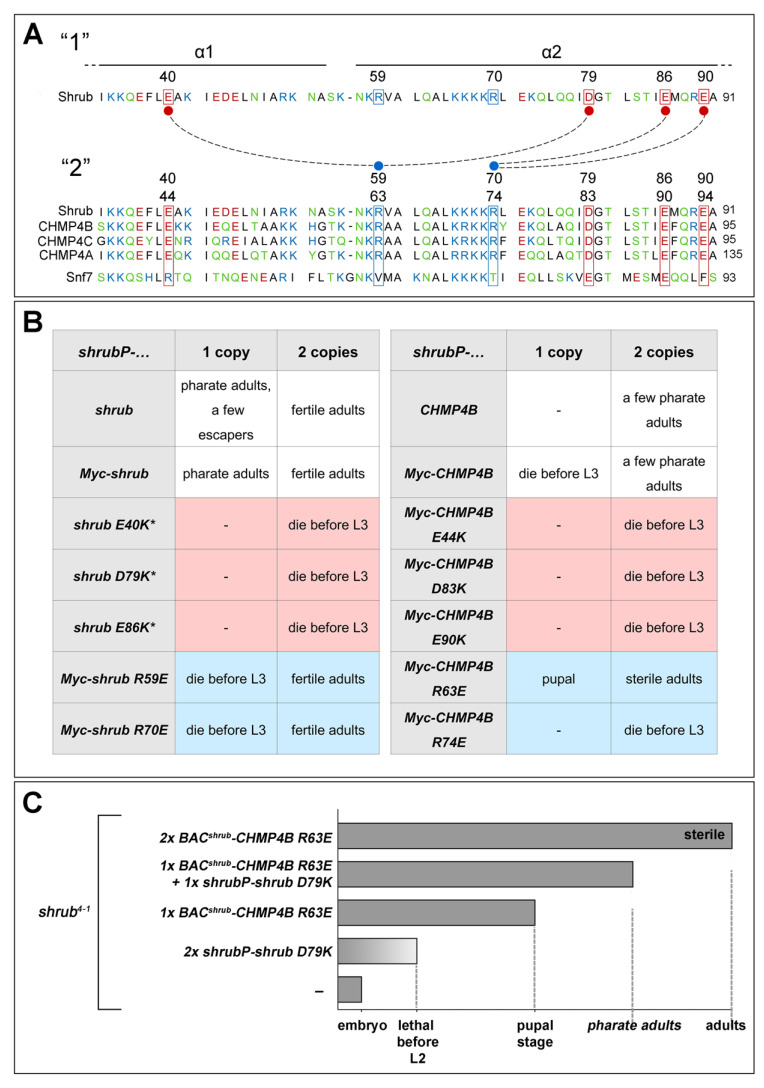
CHMP4B polymerisation relays on the same electrostatic interactions as *Drosophila* Shrub. (**A**) Primary sequence alignment of Shrub (AA 33–91), CHMP4s (AA 37–950), and yeast Snf7 (AA 34–93) (AA: polar: green; non-polar: black; acidic: red; basic: blue). Residues involved in electrostatic interactions (dotted lines) between two Shrub monomers [7] are boxed and highlighted by the residue number (upper number: Shrub, lower number: CHMP4B). (**B**) Comparison of the results from *shrub* rescue experiments based on the *shrub*^4−1^ lethality by various Shrub and human CHMP4B. (**C**) Complementation of Shrub and CHMP4B variants. A charge reversal on one position of Shrub (D79K) can be compensated by a complementary charge reversal at the position of the interacting AA in the complementary interaction surface of CHMP4B (R63E).

**Figure 3 ijms-23-07507-f003:**
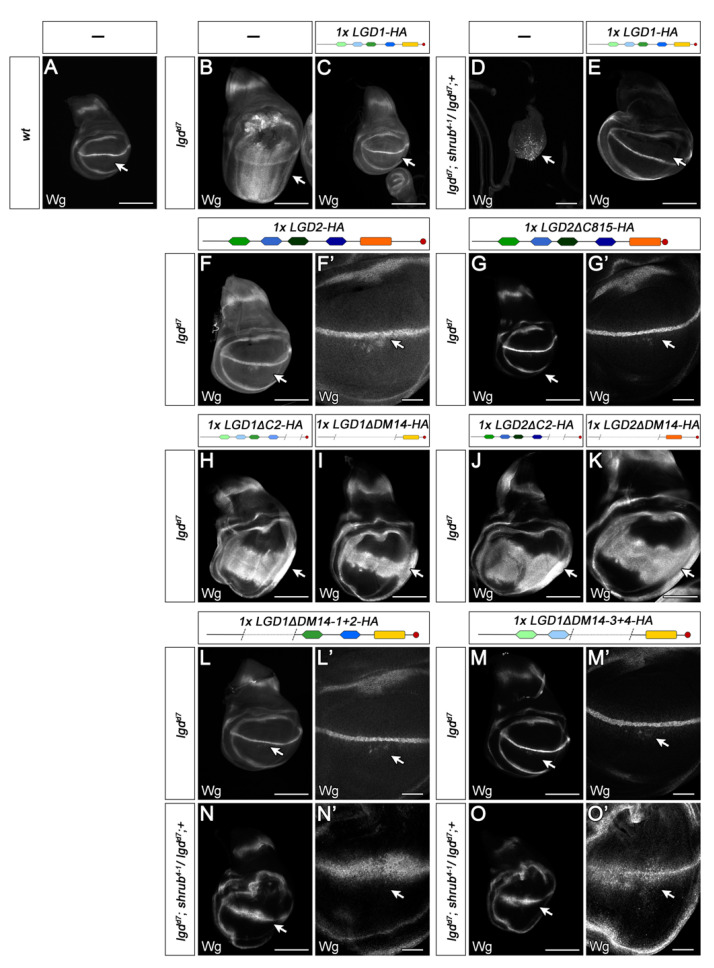
Structure/function analysis of the human LGDs in *Drosophila*. In each panel, the indicated *lgdP*-LGD1/2 variants were expressed in the *lgd* null (*lgd^d7^*) or *sensitised background* (*lgd^d7^; shrub^4−1^/lgd^d7^, +*) to assess their rescue activities. The wing imaginal discs were stained for the Notch target gene, *wingless* (*wg*) to reveal the activity of the Notch pathway (arrow). (**A**) During normal development Notch-dependent *wingless* expression is restricted to a stripe straddling the dorsal/ventral (D/V) compartment boundary. (**B**) The loss of *lgd* function causes the expansion of the expression of Wg, revealing the ectopic activation of Notch in the wing anlage. In addition, the ectopic activation of the Notch pathway also causes massive over-proliferation, leading to the increase in the size of the disc. (**D**) The phenotype of *lgd* mutant discs worsened dramatically upon loss of one copy of *shrub* (*lgd^d7^; shrub^4−1^/lgd^d7^, +*). The animals of this sensitised background die during the early third instar larval stage. They have very small discs with Wg primarily located in enlarged endosomes. The presence of one copy of *lgdP-LGD1-HA* in *lgd* mutants or the sensitised background normalises the expression of Wg and disc size indicating the rescue (**C**,**E**). (**F**,**F’**) In contrast to LGD1, LGD2 only partially rescues the *lgd* mutant phenotype, indicated by the weak ectopic expression of Wg highlighted by the arrow in (**F’**). No difference in the rescue abilities between full-length LGD2 and LGD2Δ815 have been observed (compare with **G**,**G**’). (**H**–**K**) The C2- and DM14-domains are required for the function of LGD1 (**H**,**I**) and LGD2 (**J**,**K**), indicated by the failure of rescue of the *lgd* mutant phenotype by the corresponding variants (compare with B). (**L**–**M’**) Variants of LGD1 with only one even- and odd-numbered DM14 domain (LGD1ΔDM14-1+2 and LGD1ΔDM14-3+4) can partially rescue. Only a few cells close to the D/V boundary express Wg ectopically (L’, M’, arrow). (**N**–**O’**) The ectopic expression is enhanced in the sensitised background. Scale bars (**A**–**O**) 200 µm, (**L’**–**O’**) 50 µm. At least ten wing imaginal discs were analysed for each genotype.

**Figure 4 ijms-23-07507-f004:**
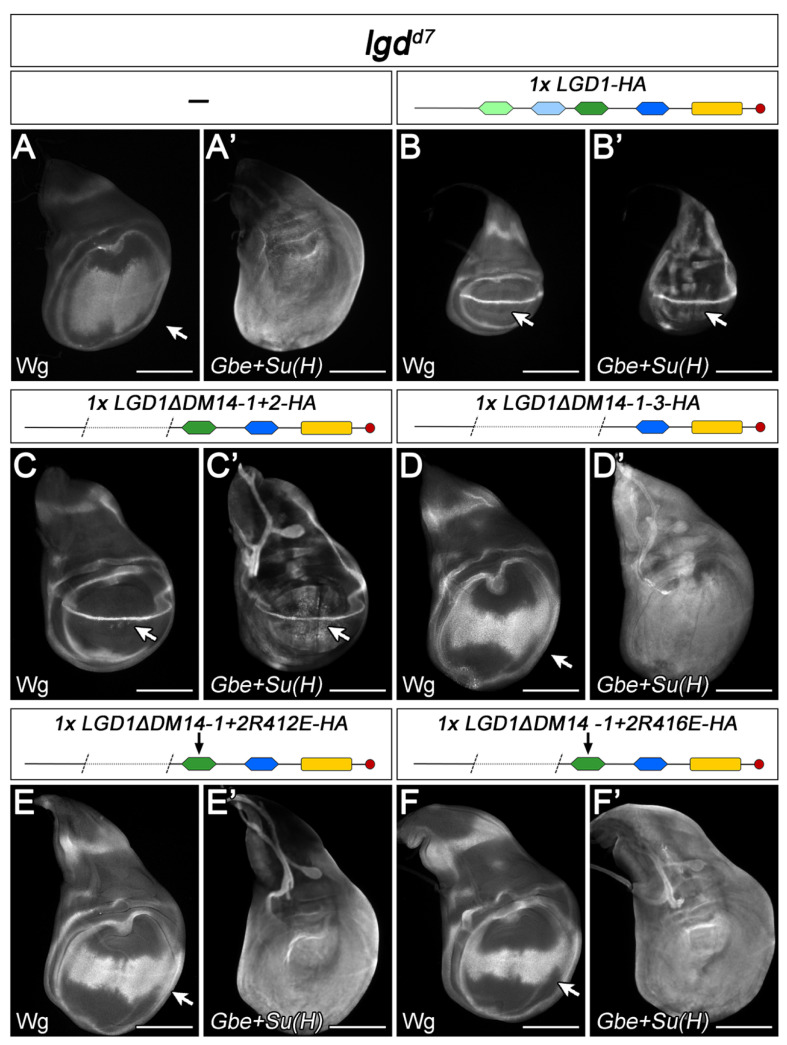
Analysis of the LGD1-Shrub interaction interface. (**A**,**A’**) Expression of Wg and the Notch activity reporter Gbe+Su(H) in *lgd* mutant discs. The expression Wg is expanded (arrow in A), while the expression of Gbe+Su(H) is expressed in all disc cells. (**B**,**B’**). The presence of one copy of lgdP-LGD1 results in a complete rescue, indicated by the normalisation of Wg expression and the pattern of Gbe+Su(H) expression. (**C**,**C’**) The presence of LGD1ΔDM14-1+2 results in a partial rescue of *lgd* mutants, indicated by the remaining weak ectopic expression of the Notch targets (arrows). (**D**,**D’**) LGD1ΔDM14-1-3 has no rescue ability, indicated by the *lgd* mutant phenotype of the discs (compare with A, A’). This shows that the odd-numbered DM14-3 is responsible for the rescue of LGD1ΔDM14-1+2. (**E**–**F’**) The rescue ability of LGD1ΔDM14-1+2 is abolished if one of the two positively charged residues within DM14-3, which mediate the interaction between protomers is exchanged to E (R412 or R416). Scale bars 200 µm. At least ten wing imaginal discs were analysed for each genotype.

**Figure 5 ijms-23-07507-f005:**
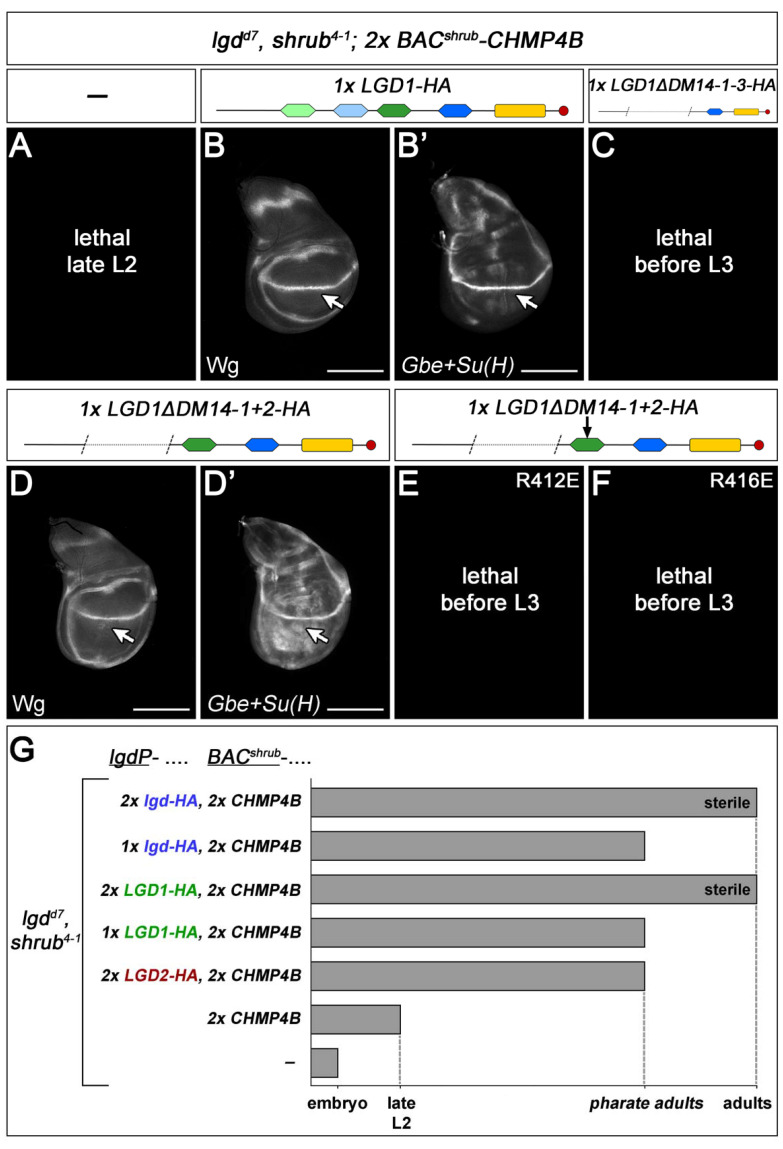
Analysis of the LGD–CHMP4B interaction. (**A**–**F**) The *lgdP*-LGD1 variants were introduced into *lgd shrub* double mutants (*lgd^d7^; shrub^4−1^)* together with two copies of *BAC^shrub^-CHMP4B*. (**A**) In the presence of two copies of *BAC^shrub^-CHMP4B*, *lgd shrub* double mutants die at the late second instar larval stage. (**B**,**G**) The additional presence of one copy of *lgdP*-LGD1-HA results in a strongly improved rescue to the pharate adult stage. In the corresponding wings discs, the expression of Wg and Gbe+Su(H) is normalised (arrow, compare with Figure 3A). (**D**,**D’**) The presence of LGD1ΔDM14-1+2 results in a weaker rescue than the full-length LGD1, indicated by the slight ectopic activation of the Notch pathway (arrow). (**C**) This rescue ability is abolished if the third DM14 is removed (LGD1ΔDM14-1-3), or if one of the two positively charged AAs within DM14-3 is exchanged (R412 or R416) (compare (**D**,**D’**) with (**E**,**F**)). (**G**) Summary of the rescue of *lgd shrub* double mutants with combinations of CHMP4B and the LGDs. CHMP4B in combination with two copies of LGD1 resulted in a full rescue of the *shrub lgd* double mutant flies, with the exception of the sterility. The combination of two copies of LGD2 and CHMP4B led to an incomplete rescue of *shrub lgd* double mutant flies to the pharate adult stage. As a control CHMP4B was also expressed with two copies of *lgdP-lgd-HA*. Scale bars 200 µm. At least ten wing imaginal discs were analysed for each genotype.

**Figure 6 ijms-23-07507-f006:**
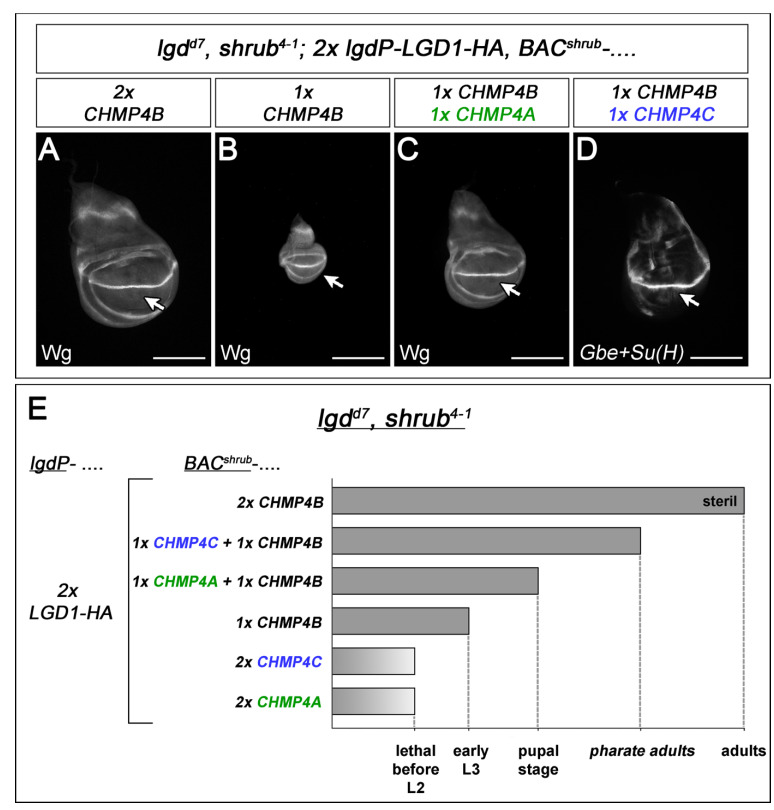
Cooperation between the CHMP4 proteins. Assay to test for the cooperation of CHMP4s is based on the finding that the combination of two copies of *lgdP*-LGD1 with one copy of *BAC^shrub^-CHMP4B* leads to a partial rescue only to the early third instar. The effects of the addition of CHMP4A and CHMP4C to this set up is tested. (**A**–**D**) The phenotype of disc of different rescue experiments are shown. (**E**) Summary of the rescue experiments. Scale bar (**A**–**D**) 200 µm. At least ten wing imaginal discs were analysed for each genotype.

## Data Availability

Not applicable.

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
