# Peer review of "Using Drosophila melanogaster to Analyse the Human Paralogs of the ESCRT-III Core Component Shrub/CHMP4/Snf7 and Its Interactions with Members of the LGD/CC2D1 Family"

_ijms, 2022, doi:10.3390/ijms23147507_

Round 1

Reviewer 1 Report

The manuscript submitted by Klien and cllueages presents a method to study the interplay between mammalian ESCRT proteins using a Drosophila model system. This work attempt to address the problem of redundancy in the ESCRT system and in fact other mammalian cellular systems that prevents pinpointing the role of each protein of the machinery. Although the idea is not novel both the methodology and results constributes to our understanding of the ESCRT machinery. Overall, I find the work interesting and solid. My main criticism is that the authors should be more careful in their conclusions because they are working with an animal model system and some of the phenotypes may be indirect or an outcome of more than one factor that has changed in the system. 

I therefore support publication after the points listed below are addressed. 

1.  I think the term humanised fly models a bit misleading. It is true that the flies were able to develop properly but the function of the human-ESCRT system was not examined per se.

2. The statement that the data indicate that CHMP4B can form functional heterofilamnets in vivo is too strong and have no direct support. you can say that the data presented supports this view but there can be other explanations such as effect on polymerization efficiency of CHMP4B by the other ESCRT-IIIs. These statements should be removed from the manuscript or toned down dramatically.

3. The limitations of this approach (indirect phenotypes for example) should be explicitly mentioned in the discussion section.

4. The abstract and introduction are too simplified, include inaccurate statements and is missing important references.

For example, VPS4 can be recruited to ESCRT-III by several components, not only CHMP2B as stated in line 55.

Also, in the abstract (line 15-16), electrostatic intearctions contribute to filamnet formation but are probably not the only one. the sentence is misleading. Also, additional references from cryo EM studies of ESCRT-III filamnets should be cited.

These are just two examples, the entire sections should be revised. Since the study focused on the mammalian ESCRTs and not on the drosophila ones, citations should include the current knowledge in the mammalian field as well.

5. figure 1: please add representative images as in the following figures.

6. Figure 1: I could not find information on the CHMP4B mutant used. This information should be included in the main text/ fig legend/ methods.

7. statistics, repetitions, and sample numbers should be added to all figures.

8. abstract and title should be revised to include the main findings of the presented work.

8. English should be corrected throughout the manuscript. There are grammatical errors, typos and more.

one example: line 274 - mis-location should be replaced with mislocalization. 

Author Response

Reviewer 1. The manuscript submitted by Klien and cllueages presents a method to study the interplay between mammalian ESCRT proteins using a Drosophila model system. This work attempt to address the problem of redundancy in the ESCRT system and in fact other mammalian cellular systems that prevents pinpointing the role of each protein of the machinery. Although the idea is not novel both the methodology and results constributes to our understanding of the ESCRT machinery. Overall, I find the work interesting and solid. My main criticism is that the authors should be more careful in their conclusions because they are working with an animal model system and some of the phenotypes may be indirect or an outcome of more than one factor that has changed in the system. 

I therefore support publication after the points listed below are addressed. 

-We thank reviewer1 for the overall very positive evaluation and the helpful suggestions for improvement. We answer the raised concern point by point in the following:

  1. I think the term humanised fly models a bit misleading. It is true that the flies were able to develop properly but the function of the human-ESCRT system was not examined per se.

-We have taken this point into account and have changed the title accordingly to: “Using Drosophila melanogaster to analyse the human paralogs of the ESCRT-III core component Shrub/CHMP4/Snf7 and its interactions with members of the LGD/CC2D1 family”

  1. The statement that the data indicate that CHMP4B can form functional heterofilamnets in vivo is too strong and have no direct support. you can say that the data presented supports this view but there can be other explanations such as effect on polymerization efficiency of CHMP4B by the other ESCRT-IIIs. These statements should be removed from the manuscript or toned down dramatically.

-we agree with the reviewer and have changed the text accordingly. We now use the term cooperate and state that the data are in agreement and suggest also an alternative in the discussion.

  1. The limitations of this approach (indirect phenotypes for example) should be explicitly mentioned in the discussion section.

-We did this by stating: “We would like to mention that while using the fly model is good as an improved cell culture system to investigate specific properties of human proteins, such as the verification/confirmation of functional interactions between two proteins and verification of important regions, the readout is indirect via the developing phenotypes and also does probably not include all aspects of functions mediated by the proteins in the human environment.”

  1. The abstract and introduction are too simplified, include inaccurate statements and is missing important references.

For example, VPS4 can be recruited to ESCRT-III by several components, not only CHMP2B as stated in line 55.

Also, in the abstract (line 15-16), electrostatic intearctions contribute to filamnet formation but are probably not the only one. the sentence is misleading. Also, additional references from cryo EM studies of ESCRT-III filamnets should be cited.

These are just two examples, the entire sections should be revised. Since the study focused on the mammalian ESCRTs and not on the drosophila ones, citations should include the current knowledge in the mammalian field as well.

-we have re-written the abstract and Introduction in line with the suggestions.

  1. figure 1: please add representative images as in the following figures.

-done

  1. Figure 1: I could not find information on the CHMP4B mutant used. This information should be included in the main text/ fig legend/ methods.

-we have added the necessary information

  1. statistics, repetitions, and sample numbers should be added to all figures.

-done

  1. abstract and title should be revised to include the main findings of the presented work.

-done

  1. English should be corrected throughout the manuscript. There are grammatical errors, typos and more.

one example: line 274 - mis-location should be replaced with mislocalization. 

-we tried to improve..

Reviewer 2: The work is very well written and presented. The introduction is concise and descriptive enough. the results are clear and revealing, although I would have liked to see some of the supplementary figures in the main article. The article may be published in its present form.

Reviewer 2 Report

The work is very well written and presented. The introduction is concise and descriptive enough. the results are clear and revealing, although I would have liked to see some of the supplementary figures in the main article. The article may be published in its present form.

Author Response

Reviewer 2: The work is very well written and presented. The introduction is concise and descriptive enough. the results are clear and revealing, although I would have liked to see some of the supplementary figures in the main article. The article may be published in its present form.

-we thank reviewer2 for the very nice evaluation.

Reviewer 3 Report

In their manuscript titled "Using a humanized fly model to analyze the function of the human paralogs of the ESCRT-III core component Shrub/CHMP4/Snf7 and its interactions with the members of the LGD/CC2D1 family" the authors analyzed the function of the human orthologs of Shrub and Lgd in the fly model Drosophila melanogaster by taking advantage of its genetics versatility. The Shrub/CHMP4s, and its regulator CC2D1/LGD, proteins are at the heart of the Endosomal Sorting Complex Required for Transport III (ESCRT-III) which is well-conserved machinery required to pinch off membrane patches from different sub-cellular compartments. Though, the authors state that they used the fly model because enabling them"... to use a plethora of available techniques..." they mostly stuck to genetics, ignoring biochemical assays. Overall, the manuscript presents interesting insights, though prior publication few inaccuracies should be amended as below shortly summarized.  

  1. Referring to the first part of the section "Results" (lines 98-99 and 100-101) the authors should detail a little bit more a couple of points. a) "...we tested the functionality of various tagged variants...". How many? How did they were constructed? Where do they come from? Are those depicted ie Fig. S1? If yest it should be mentioned; b) when it comes to the lines 100-101 the authors stated"...shrub promoter fragment encompassing 500 bp up- and down-stream of the shrub ORF...". When a sequence is downstream the ORF it should not be considered terminator rather than promoter. Is it not? Please rephrase the sentence;
  2. Figure 1: since there is just a single panel please remove the letter A. In addition, referring to such rescue experiments please implement the Supplementary Material by adding a Figure in which pictures of the different viable strains are shown. This should apply also to the others rescue assays;
  3. The Fig. 3A refers to the w.t. strain and not to the lgd loss of function strain (panel B) as indicated within the main text (lines 230-232). In addition, Fig. 3 and its legend should be revised because the version I have got lacks of panel O (line 259);
  4. What do the authors mean for normalized (line 363)? Normalized against what?
  5. Throughout the manuscript, the authors refer to protein-protein interaction occurring between LGD/CC2D1 and CHMP4 (e.g. lines 389-390), though they came to such conclusions by means of genetics, and not by biochemical, approach. For the sake of clarity in the main text this issue should be clarified, at least with a sentence, just to dispel a kind of confusion due to the fact that CC2D1 might also play potential roles that are ESCRT-III independent;
  6. In the version of the manuscript that I downloaded panels B and C of Figure 6 are missing. Thus they should be provided. Otherwise, the Figure should be edited accordingly;
  7. Materials and Methods should be detailed a little bit more when it comes to plasmids construction. Consistently, please provide the complete list of the primers used as a Supplementary Table and detail a little bit more the construction of the plasmids (e.g human CHMP4s cDNA have been isolated from RNA coming from which tissue? with which primers? etc...). Furthermore, what does NGS stand for? Please indicate the catalog number for the protease inhibitors from SIGMA and the detail about the antibody used to probe the membrane (Fig. S6) with tubulin.
  8. Fig S6 panels B and B' please provide a shorter exposure because tubulin bands result overexposed when compared to CC2D1A and HA;
  9. The manuscript formatting requires to be revised by numbering the different sections as indicated by the journal Editorial policy;
  10. A few typos are present throughout the manuscript and they should be edited (e.g. lines 236-237, 433, 615, 644).

Author Response

Reviewer3:

-we thank reviewer3 for the positive evaluation. However, we could not explicitly take all advices for improvement in account due to the fact that the review has not been sent to us and we have discovered the delayed review only at the time of re-submission. However, we realized that we have addressed a lot of the concerns, since they were similar to that of one other reviewer. We had only time to rephrase some sentences along the lines suggested.

  1. Referring to the first part of the section "Results" (lines 98-99 and 100-101) the authors should detail a little bit more a couple of points. a) "...we tested the functionality of various tagged variants...". How many? How did they were constructed? Where do they come from? Are those depicted ie Fig. S1? If yest it should be mentioned; b) when it comes to the lines 100-101 the authors stated"...shrub promoter fragment encompassing 500 bp up- and down-stream of the shrub ORF...". When a sequence is downstream the ORF it should not be considered terminator rather than promoter. Is it not? Please rephrase the sentence;

-done.

  1. Figure 1: since there is just a single panel please remove the letter A. In addition, referring to such rescue experiments please implement the Supplementary Material by adding a Figure in which pictures of the different viable strains are shown. This should apply also to the others rescue assays;

-done

  1. The Fig. 3A refers to the w.t. strain and not to the lgd loss of function strain (panel B) as indicated within the main text (lines 230-232). In addition, Fig. 3 and its legend should be revised because the version I have got lacks of panel O (line 259);

-done

  1. What do the authors mean for normalized (line 363)? Normalized against what?

-the expression of the target gene has normalised, meaning it has a wildtype expression. This is a phrase commonly used in Drosophila field describing rescue assays.

  1. Throughout the manuscript, the authors refer to protein-protein interaction occurring between LGD/CC2D1 and CHMP4 (e.g. lines 389-390), though they came to such conclusions by means of genetics, and not by biochemical, approach. For the sake of clarity in the main text this issue should be clarified, at least with a sentence, just to dispel a kind of confusion due to the fact that CC2D1 might also play potential roles that are ESCRT-III independent;

  1. In the version of the manuscript that I downloaded panels B and C of Figure 6 are missing. Thus they should be provided. Otherwise, the Figure should be edited accordingly;

-done

  1. Materials and Methods should be detailed a little bit more when it comes to plasmids construction. Consistently, please provide the complete list of the primers used as a Supplementary Table and detail a little bit more the construction of the plasmids (e.g human CHMP4s cDNA have been isolated from RNA coming from which tissue? with which primers? etc...). Furthermore, what does NGS stand for? Please indicate the catalog number for the protease inhibitors from SIGMA and the detail about the antibody used to probe the membrane (Fig. S6) with tubulin.
  2. Fig S6 panels B and B' please provide a shorter exposure because tubulin bands result overexposed when compared to CC2D1A and HA;
  3. The manuscript formatting requires to be revised by numbering the different sections as indicated by the journal Editorial policy;
  4. A few typos are present throughout the manuscript and they should be edited (e.g. lines 236-237, 433, 615, 644).

-hopefully done